# MCVD: Masked Conditional Video Diffusion for Prediction, Generation, and Interpolation

**Vikram Voleti**[*]
Mila, University of Montreal
Canada
vikram.voleti@umontreal.ca

**Alexia Jolicoeur-Martineau**[*]
Mila, University of Montreal
Canada
alexia.jolicoeur-martineau@mail.mcgill.ca

**Christopher Pal**
Mila, Polytechnique Montreal
Canada CIFAR AI Chair
ServiceNow Research

## Abstract

Video prediction is a challenging task. The quality of video frames from current state-of-the-art (SOTA) generative models tends to be poor and generalization beyond the training data is difficult. Furthermore, existing prediction frameworks are typically not capable of simultaneously handling other video-related tasks such as unconditional generation or interpolation. In this work, we devise a general-purpose framework called Masked Conditional Video Diffusion (MCVD) for all of these video synthesis tasks using a probabilistic conditional score-based denoising diffusion model, conditioned on past and/or future frames. We train the model in a manner where we randomly and independently mask all the past frames or all the future frames. This novel but straightforward setup allows us to train a single model that is capable of executing a broad range of video tasks, specifically: future/past prediction – when only future/past frames are masked; unconditional generation – when both past and future frames are masked; and interpolation – when neither past nor future frames are masked. Our experiments show that this approach can generate high-quality frames for diverse types of videos. Our MCVD models are built from simple non-recurrent 2D-convolutional architectures, conditioning on blocks of frames and generating blocks of frames. We generate videos of arbitrary lengths autoregressively in a block-wise manner. Our approach yields SOTA results across standard video prediction and interpolation benchmarks, with computation times for training models measured in 1-12 days using $\leq 4$ GPUs.

Project page: https://mask-cond-video-diffusion.github.io
Code: https://mask-cond-video-diffusion.github.io/

## 1 Introduction

Predicting what one may visually perceive in the future is closely linked to the dynamics of objects and people. As such, this kind of prediction relates to many crucial human decision-making tasks ranging from making dinner to driving a car. If video models could generate full-fledged videos in pixel-level detail with plausible futures, agents could use them to make better decisions, especially safety-critical ones. Consider, for example, the task of driving a car in a tight situation at high speed. Having an accurate model of the future could mean the difference between damaging a car or

---

[*]Equal Contribution

36th Conference on Neural Information Processing Systems (NeurIPS 2022).

something worse. We can obtain some intuitions about this scenario by examining the predictions of our model in Figure 1, where we condition on two frames and predict 28 frames into the future for a car driving around a corner. We can see that this is enough time for two different painted arrows to pass under the car. If one zooms in, one can inspect the relative positions of the arrow and the Mercedes hood ornament in the real versus predicted frames. Pixel-level models of trajectories, pedestrians, potholes, and debris on the road could one day improve the safety of vehicles.

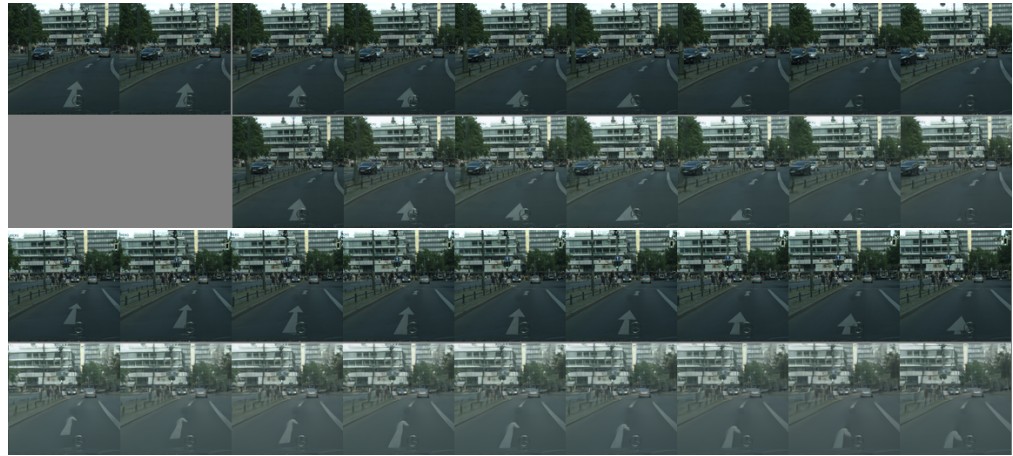

Figure 1: Our approach generates high quality frames many steps into the future: Given two conditioning frames from the Cityscapes [Cordts et al., 2016] validation set (top left), we show 7 predicted future frames in row 2 below, then skip to frames 20-28, autoregressively predicted in row 4. Ground truth frames are shown in rows 1 and 3. Notice the initial large arrow advancing and passing under the car. In frame 20 (the far left of the 3rd and 4th row), the initially small and barely visible second arrow in the background of the conditioning frames has advanced into the foreground. Result generated by our **MCVD** concat model variant. Note that some Cityscapes videos contain brightness changes, which may explain the brightness change in this sample.

Although beneficial to decision making, video generation is an incredibly challenging problem; not only must high-quality frames be generated, but the changes over time must be plausible and ideally drawn from an accurate and potentially complex distribution over probable futures. Looking far in time is exceptionally hard given the exponential increase in possible futures. Generating video from scratch or unconditionally further compounds the problem because even the structure of the first frame must be synthesized. Also related to video generation are the simpler tasks of a) video prediction, predicting the future given the past, and b) interpolation, predicting the in-between given past and future. Yet, both problems remain challenging. Specialized tools exist to solve the various video tasks, but they rarely solve more than one task at a time.

Given the monumental task of general video generation, current approaches are still very limited despite the fact that many state of the art methods have hundreds of millions of parameters [Wu et al., 2021, Weissenborn et al., 2019, Villegas et al., 2019, Babaeizadeh et al., 2021]. While industrial research is capable of looking at even larger models, current methods frequently underfit the data, leading to blurry videos, especially in the longer-term future and recent work has examined ways in improve parameter efficiency [Babaeizadeh et al., 2021]. Our objective here is to devise a video generation approach that generates high-quality, time-consistent videos within our computation budget of $\leq 4$ GPU) and computation times for training models $\leq$ two weeks. Fortunately, diffusion models for image synthesis have demonstrated wide success, which strongly motivated our use of this approach. Our qualitative results in Figure 1 also indicate that our particular approach does quite well at synthesizing frames in the longer-term future (i.e., frame 29 in the bottom right corner).

One family of diffusion models might be characterized as Denoising Diffusion Probabilistic Models (DDPMs) [Sohl-Dickstein et al., 2015, Ho et al., 2020, Dhariwal and Nichol, 2021], while another as Score-based Generative Models (SGMs) [Song and Ermon, 2019, Li et al., 2019, Song and Ermon, 2020, Jolicoeur-Martineau et al., 2021a]. However, these approaches have effectively merged into a field we shall refer to as score-based diffusion models, which work by defining a stochastic process from data to noise and then reversing that process to go from noise to data. Their main benefits are that they generate very 1) high-quality and 2) diverse data samples. One of their drawbacks is that solving the reverse process is relatively slow, but there are ways to improve speed [Song et al., 2020,

Jolicoeur-Martineau et al., 2021b, Salimans and Ho, 2022, Liu et al., 2022, Xiao et al., 2022]. Given their massive success and attractive properties, we focus here on developing our framework using score-based diffusion models for video prediction, generation, and interpolation.

Our work makes the following contributions:

1. A conditional video diffusion approach for video prediction and interpolation that yields SOTA results.

2. A conditioning procedure based on masking past and/or future frames in a blockwise manner giving a single model the ability to solve multiple video tasks: future/past prediction, unconditional generation, and interpolation.

3. A sliding window *blockwise autoregressive* conditioning procedure to allow fast and coherent long-term generation (Figure 2).

4. A convolutional U-net neural architecture integrating recent developments with a conditional normalization technique we call SPAce-TIme-Adaptive Normalization (SPATIN) (Figure 3).

By conditioning on blocks of frames in the past and optionally blocks of frames even further in the future, we are able to better ensure that temporal dynamics are transferred across blocks of samples, i.e. our networks can learn *implicit* models of spatio-temporal dynamics to inform frame generation. Unlike many other approaches, we do not have explicit model components for spatio-temporal derivatives or optical flow or recurrent blocks.

## 2 Conditional Diffusion for Video

Let $\mathbf{x}_0 \in \mathbb{R}^d$ be a sample from the data distribution $p_{\text{data}}$. A sample $\mathbf{x}_0$ can corrupted from $t = 0$ to $t = T$ through the Forward Diffusion Process (FDP) with the following transition kernel:

$$q_t(\mathbf{x}_t|\mathbf{x}_{t-1}) = \mathcal{N}(\mathbf{x}_t; \sqrt{1-\beta_t}\mathbf{x}_{t-1}, \beta_t\mathbf{I}), \tag{1}$$

Furthermore, $\mathbf{x}_t$ can be sampled directly from $\mathbf{x}_0$ using the following accumulated kernel:

$$q_t(\mathbf{x}_t|\mathbf{x}_0) = \mathcal{N}(\mathbf{x}_t; \sqrt{\bar{\alpha}_t}\mathbf{x}_0, (1-\bar{\alpha}_t)\mathbf{I}) \implies \mathbf{x}_t = \sqrt{\bar{\alpha}_t}\mathbf{x}_0 + \sqrt{1-\bar{\alpha}_t}\boldsymbol{\epsilon} \tag{2}$$

where $\bar{\alpha}_t = \prod_{s=1}^t (1-\beta_s)$, and $\boldsymbol{\epsilon} \sim \mathcal{N}(\mathbf{0}, \mathbf{I})$.

Generating new samples can be done by reversing the FDP and solving the Reverse Diffusion Process (RDP) starting from Gaussian noise $\mathbf{x}_T$. It can be shown (Song et al. [2021], Ho et al. [2020]) that the RDP can be computed using the following transition kernel:

$$p_t(\mathbf{x}_{t-1}|\mathbf{x}_t, \mathbf{x}_0) = \mathcal{N}(\mathbf{x}_{t-1}; \tilde{\boldsymbol{\mu}}_t(\mathbf{x}_t, \mathbf{x}_0), \tilde{\beta}_t\mathbf{I}),$$

$$\text{where} \quad \tilde{\boldsymbol{\mu}}_t(\mathbf{x}_t, \mathbf{x}_0) = \frac{\sqrt{\bar{\alpha}_{t-1}}\beta_t}{1-\bar{\alpha}_t}\mathbf{x}_0 + \frac{\sqrt{\alpha_t}(1-\bar{\alpha}_{t-1})}{1-\bar{\alpha}_t}\mathbf{x}_t \quad \text{and} \quad \tilde{\beta}_t = \frac{1-\bar{\alpha}_{t-1}}{1-\bar{\alpha}_t}\beta_t \tag{3}$$

Since $\mathbf{x}_0$ given $\mathbf{x}_t$ is unknown, it can be estimated using eq. (2): $\hat{\mathbf{x}}_0 = (\mathbf{x}_t - \sqrt{1-\bar{\alpha}_t}\boldsymbol{\epsilon})/\sqrt{\bar{\alpha}_t}$, where $\boldsymbol{\epsilon}_\theta(\mathbf{x}_t|t)$ estimates $\boldsymbol{\epsilon}$ using a time-conditional neural network parameterized by $\theta$. This allows us to reverse the process from noise to data. The loss function of the neural network is:

$$L(\theta) = \mathbb{E}_{t, \mathbf{x}_0 \sim p_{\text{data}}, \boldsymbol{\epsilon} \sim \mathcal{N}(\mathbf{0},\mathbf{I})} \left[ \left\| \boldsymbol{\epsilon} - \boldsymbol{\epsilon}_\theta(\sqrt{\bar{\alpha}_t}\mathbf{x}_0 + \sqrt{1-\bar{\alpha}_t}\boldsymbol{\epsilon} \mid t) \right\|_2^2 \right] \tag{4}$$

Note that estimating $\boldsymbol{\epsilon}$ is equivalent to estimating a scaled version of the score function (i.e., the gradient of the log density) of the noisy data:

$$\nabla_{\mathbf{x}_t} \log q_t(\mathbf{x}_t \mid \mathbf{x}_0) = -\frac{1}{1-\bar{\alpha}_t}(\mathbf{x}_t - \sqrt{\bar{\alpha}_t}\mathbf{x}_0) = -\frac{1}{\sqrt{1-\bar{\alpha}_t}}\boldsymbol{\epsilon} \tag{5}$$

Thus, data generation through denoising depends on the score-function, and can be seen as noise-conditional score-based generation.

Score-based diffusion models can be straightforwardly adapted to video by considering the joint distribution of multiple continuous frames. While this is sufficient for unconditional video generation, other tasks such as video interpolation and prediction remain unsolved. A conditional video prediction model can be approximately derived from the unconditional model using imputation [Song et al., 2021]; indeed, the contemporary work of Ho et al. [2022] attempts to use this technique; however, their approach is based on an approximate conditional model.

## 2.1 Video Prediction via Conditional Diffusion

We first propose to directly model the conditional distribution of video frames in the immediate future given past frames. Assume we have $p$ past frames $\mathbf{p} = \left\{\mathbf{p}^i\right\}_{i=1}^p$ and $k$ current frames in the immediate future $\mathbf{x}_0 = \left\{\mathbf{x}_0^i\right\}_{i=1}^k$. We condition the above diffusion models on the past frames to predict the current frames:

$$L_{\text{vidpred}}(\theta) = \mathbb{E}_{t,[\mathbf{p},\mathbf{x}_0]\sim p_{\text{data}},\boldsymbol{\epsilon}\sim\mathcal{N}(\mathbf{0},\mathbf{I})}\left[\left\|\boldsymbol{\epsilon} - \boldsymbol{\epsilon}_\theta(\sqrt{\bar{\alpha}_t}\mathbf{x}_0 + \sqrt{1-\bar{\alpha}_t}\boldsymbol{\epsilon} \mid \mathbf{p}, t)\right\|^2\right] \quad (6)$$

Given a model trained as above, video prediction for subsequent time steps can be achieved by blockwise autoregressively predicting current video frames conditioned on previously predicted frames (see Figure 2). We use variants of the network shown in Figure 3 to model $\boldsymbol{\epsilon}_\theta$ in Equation 6 here, and for Equation 7 and Equation 8 below.

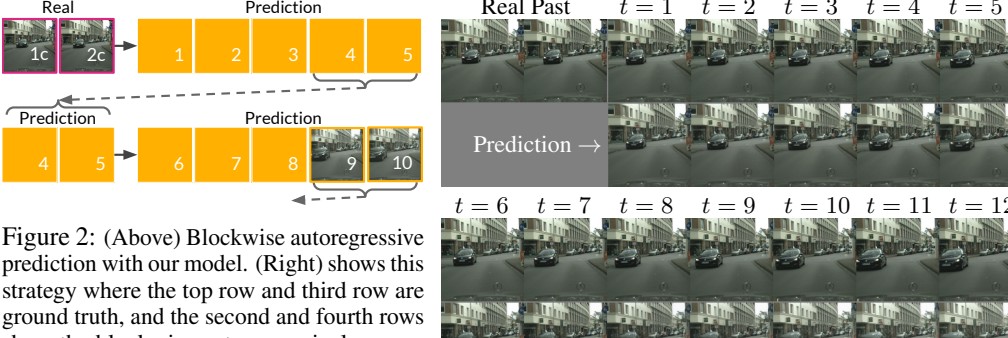

Figure 2: (Above) Blockwise autoregressive prediction with our model. (Right) shows this strategy where the top row and third row are ground truth, and the second and fourth rows show the blockwise autoregressively generated frames using our approach.

## 2.2 Video Prediction + Generation via Masked Conditional Diffusion

Our approach above allows video prediction, but not unconditional video generation. As a second approach, we extend the same framework to video generation by masking (zeroing-out) the past frames with probability $p_{\text{mask}} = 1/2$ using binary mask $m_p$. The network thus learns to predict the noise added without any past frames for context. Doing so means that we can perform conditional as well as unconditional frame generation, i.e., video prediction and generation with the same network. This leads to the following loss ($\mathcal{B}$ is the Bernouilli distribution):

$$L_{\text{vidgen}}(\theta) = \mathbb{E}_{t,[\mathbf{p},\mathbf{x}_0]\sim p_{\text{data}},\boldsymbol{\epsilon}\sim\mathcal{N}(\mathbf{0},\mathbf{I}),m_p\sim\mathcal{B}(p_{\text{mask}})}\left[\left\|\boldsymbol{\epsilon} - \boldsymbol{\epsilon}_\theta(\sqrt{\bar{\alpha}_t}\mathbf{x}_0 + \sqrt{1-\bar{\alpha}_t}\boldsymbol{\epsilon} \mid m_p\mathbf{p}, t)\right\|^2\right] \quad (7)$$

We hypothesize that this dropout-like [Srivastava et al., 2014] approach will also serve as a form of regularization, improving the model's ability to perform predictions conditioned on the past. We see positive evidence of this effect in our experiments – see the MCVD past-mask model variants in Tables 3 and 9 versus without past-masking. Note that random masking is used only during training.

## 2.3 Video Prediction + Generation + Interpolation via Masked Conditional Diffusion

We now have a design for video prediction and generation, but it still cannot perform video interpolation nor past prediction from the future. As a third and final approach, we show how to build a general model for solving all four video tasks. Assume we have $p$ past frames, $k$ current frames, and $f$ future frames $\mathbf{f} = \left\{\mathbf{f}^i\right\}_{i=1}^f$. We randomly mask the $p$ past frames with probability $p_{mask} = 1/2$, and similarly randomly mask the $f$ future frames with the same probability (but sampled separately). Thus, future or past prediction is when only future or past frames are masked. Unconditional generation is when both past and future frames are masked. Video interpolation is when neither past nor future frames are masked. The loss function for this general video machinery is:

$$L(\theta) = \mathbb{E}_{t,[\mathbf{p},\mathbf{x}_0,\mathbf{f}]\sim p_{\text{data}},\boldsymbol{\epsilon}\sim\mathcal{N}(\mathbf{0},\mathbf{I}),(m_p,m_f)\sim\mathcal{B}(p_{\text{mask}})}\left[\left\|\boldsymbol{\epsilon} - \boldsymbol{\epsilon}_\theta(\sqrt{\bar{\alpha}_t}\mathbf{x}_0 + \sqrt{1-\bar{\alpha}_t}\boldsymbol{\epsilon} \mid m_p\mathbf{p}, m_f\mathbf{f}, t)\right\|^2\right]$$

$$(8)$$

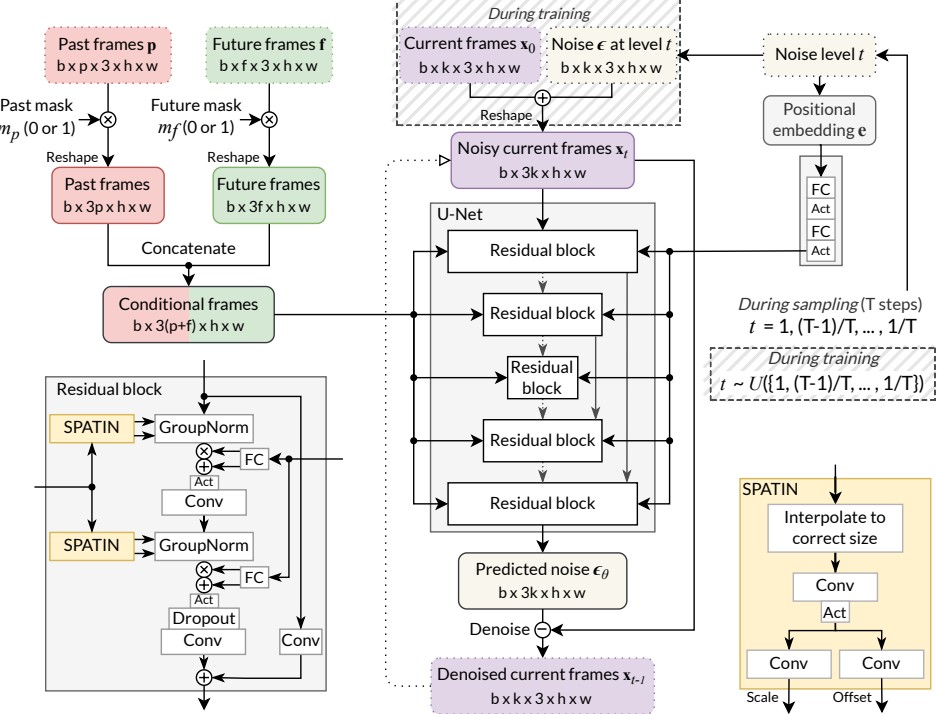

Figure 3: We give noisy current frames to a U-Net whose residual blocks receive conditional information from past/future frames and noise-level. The output is the predicted noise in the current frames, which we use to denoise the current frames. At test time, we start from pure noise.

## 2.4 Our Network Architecture

For our denoising network we use a U-net architecture [Ronneberger et al., 2015, Honari et al., 2016, Salimans et al., 2017] combining the improvements from Song et al. [2021] and Dhariwal and Nichol [2021]. This architecture uses a mix of 2D convolutions [Fukushima and Miyake, 1982], multi-head self-attention [Cheng et al., 2016], and adaptive group-norm [Wu and He, 2018]. We use positional encodings of the noise level ($t \in [0, 1]$) and process it using a transformer style positional embedding:

$$\mathbf{e}(t) = \left[\ldots, \cos\left(tc^{\frac{-2d}{D}}\right), \sin\left(tc^{\frac{-2d}{D}}\right), \ldots\right]^{\mathrm{T}},\qquad(9)$$

where $d = 1, \ldots, D/2$, $D$ is the number of dimensions of the embedding, and $c = 10000$. This embedding vector is passed through a fully connected layer, followed by an activation function and another fully connected layer. Each residual block has an fully connected layer that adapts the embedding to the correct dimensionality.

To provide $\mathbf{x}_t$, $\mathbf{p}$, and $\mathbf{f}$ to the network, we separately concatenate the past/future conditional frames and the noisy current frames in the channel dimension. The concatenated noisy current frames are directly passed as input to the network. Meanwhile, the concatenated conditional frames are passed through an embedding that influences the conditional normalization akin to SPatially-Adaptive (DE)normalization (SPADE) [Park et al., 2019]; to account for the effect of time/motion, we call this approach SPAce-TIme-Adaptive Normalization (SPATIN). In addition to SPATIN, we also try directly concatenating the conditional and noisy current frames together and passing them as the input. In our experiments below we show some results with SPATIN and some with concatenation (concat). For simple video prediction with Equation 6, we experimented with 3D convolutions and 3D attention However, this requires an exorbitant amount of memory, and we found no benefit in using 3D layers over 2D layers at the same memory (i.e., the biggest model that fits in 4 GPUs). Thus, we did not explore this idea further. We also tried and found no benefit from gamma noise [Nachmani et al., 2021], L1 loss, and F-PNDM sampling [Liu et al., 2022].

# 3 Related work

Score-based diffusion models have been used for image editing [Meng et al., 2022, Saharia et al., 2021, Nichol et al., 2021] and our approach to video generation might be viewed as an analogy to classical image inpainting, but in the temporal dimension. The GLIDE or Guided Language to Image Diffusion for Generation and Editing approach of Nichol et al. [2021] uses CLIP-guided diffusion for image editing, while Denoising Diffusion Restoration Models (DDRM) Kawar et al. [2022] additionally condition on a corrupted image to restore the clean image. Adversarial variants of score-based diffusion models have been used to enhance quality [Jolicoeur-Martineau et al., 2021a] or speed [Xiao et al., 2022].

Contemporary work to our own such as that of Ho et al. [2022] and Yang et al. [2022] also examine video generation using score-based diffusion models. However, the Video Diffusion Models (VDMs) work of Ho et al. [2022] approximates conditional distributions using a gradient method for conditional sampling from their unconditional model formulation. In contrast, our approach directly works with a conditional diffusion model, which we obtain through masked conditional training, thereby giving us the exact conditional distribution as well as the ability to generate unconditionally. Their experiments focus on: a) unconditional video generation, and b) text-conditioned video generation, whereas our work focuses primarily on predicting future video frames from the past, using our masked conditional generation framework. The Residual Video Diffusion (RVD) of Yang et al. [2022] is only for video prediction, and it uses a residual formulation to generate frames autoregressively one at a time. Meanwhile, ours directly models the conditional frames to generate multiple frames in a block-wise autoregressive manner.

Recurrent neural network (RNN) techniques were early candidates for modern deep neural architectures for video prediction and generation. Early work combined RNNs with a stochastic latent variable (SV2P) Babaeizadeh et al. [2018a] and was optimized by variational inference. The stochastic video generation (SVG) approach of Denton and Fergus [2018] learned both prior and a per time step latent variable model, which influences the dynamics of an LSTM at each step. The model is also trained in a manner similar to a variational autoencoder, i.e., it was another form of variational RNN (vRNN). To address the fact that vRNNs tend to lead to blurry results, Castrejón et al. [2019] (Hier-vRNN) increased the expressiveness of the latent distributions using a hierarchy of latent variables. We compare qualitative result of SVG and Hier-vRNN with the **MCVD** concat variant of our method in Figure 4. Other vRNN-based models include SAVP Lee et al. [2018], SRVP Franceschi et al. [2020], SLAMP Akan et al. [2021].

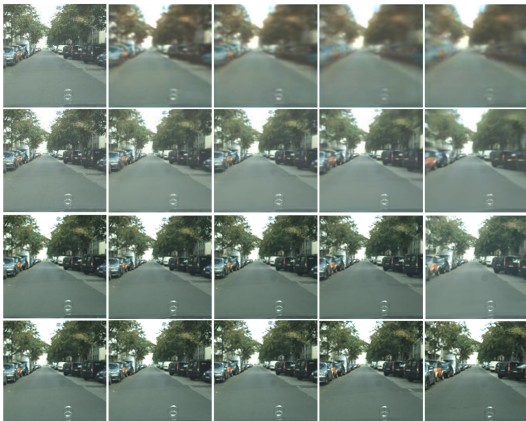

Figure 4: Comparing future prediction methods on Cityscapes: SVG-LP (Top Row), Hier-vRNNs (Second Row), Our Method (Third Row), Ground Truth (Bottom Row). Frame 2, a ground truth conditioning frame is shown in first column, followed by frames: 3, 5, 10 and 20 generated by each method vs the ground truth at the bottom.

The well known Transformer paradigm [Vaswani et al., 2017] from natural language processing has also been explored for video. The Video-GPT work of Yan et al. [2021] applied an autoregressive GPT style [Brown et al., 2020] transformer to the codes produced from a VQ-VAE [Van Den Oord et al., 2017]. The Video Transformer work of Weissenborn et al. [2019] models video using 3-D spatio-temporal volumes without linearizing positions in the volume. They examine local self-attention over small non-overlapping sub-volumes or 3D blocks. This is done partly to accelerate computations on TPU hardware. Their work also observed that the peak signal-to-noise ratio (PSNR) metric and the mean-structural similarity (SSIM) metrics [Wang et al., 2004] were developed for images, and have serious flaws when applied to videos. PSNR prefers blurry videos and SSIM does not correlate well to perceptual quality. Like them, we focus on the recently proposed Frechet Video Distance (FVD) [Unterthiner et al., 2018], computed over entire videos and which is sensitive to visual quality, temporal coherence, and diversity of samples. Rakhimov et al. [2020] (LVT) used transformers to

predict the dynamics of video in latent space. Le Moing et al. [2021] (CCVS) also predict in latent space, that of an adversarially trained autoencoder, and also add a learnable optical flow module.

Generative Adversarial Network (GAN) based approaches to video generation have also been studied extensively. Vondrick et al. [2016] proposed an early GAN architecture for video, using a spatio-temporal CNN. Villegas et al. [2017] proposed a strategy for separating motion and content into different pathways of a convolutional LSTM based encoder-decoder RNN. Saito et al. [2017] (TGAN) predicted a sequence of latents using a temporal generator, and then the sequence of frames from those latents using an image generator. TGANv2 Saito et al. [2020] improved its memory efficiency. MoCoGAN Tulyakov et al. [2018] explored style and content separation, but within a CNN framework. Yushchenko et al. [2019] used the MoCoGAN framework by re-formulating the video prediction problem as a Markov Decision Process (MDP). FutureGAN Aigner and Körner [2018] used spatio-temporal 3D convolutions in an encoder decoder architecture, and elements of the progressive GAN Karras et al. [2018] approach to improve image quality. TS-GAN Munoz et al. [2021] facilitated information flow between consecutive frames. TriVD-GAN Luc et al. [2020] proposes a novel recurrent unit in the generator to handle more complex dynamics, while DIGAN Yu et al. [2022] uses implicit neural representations in the generator.

Video interpolation was the subject of a flurry of interest in the deep learning community a number of years ago [Niklaus et al., 2017, Jiang et al., 2018, Xue et al., 2019, Bao et al., 2019]. However, these architectures tend to be fairly specialized to the interpolation task, involving optical flow or motion field modelling and computations. Frame interpolation is useful for video compression; therefore, many other lines of work have examined interpolation from a compression perspective. However, these architectures tend to be extremely specialized to the video compression task [Yang et al., 2020].

The Cutout approach of DeVries and Taylor [2017] has examined the idea of cutting out small continuous regions of an input image, such as small squares. Dropout [Srivastava et al., 2014] at the FeatureMap level was proposed and explored under the name of SpatialDropout in Tompson et al. [2015]. Input Dropout [de Blois et al., 2020] has been examined in the context of dropping different channels of multi-modal input imagery, such as the dropping of the RGB channels or depth map channels during training, then using the model without one of the modalities during testing, e.g. in their work they drop the depth channel.

Regarding our block-autoregressive approach, previous video prediction models were typically either 1) non-recurrent: predicting all $n$ frames simultaneously with no way of adding more frames (most GAN-based methods), or 2) recurrent in nature, predicting 1 frame at a time in an autoregressive fashion. The benefit of the non-recurrent type is that you can generate videos faster than 1 frame at a time while allowing for generating as many frames as needed. The disadvantage is that it is slower than generating all frames at once, and takes up more memory and compute at each iteration. Our model finds a sweet spot in between in that it is block-autoregressive: generating $k < n$ frames at a time recurrently to finally obtain $n$ frames.

# 4 Experiments

We show the results of our **video prediction** experiments on test data that was never seen during training in Tables 1 - 4 for Stochastic Moving MNIST (SMMNIST) [2], KTH [3], BAIR [4], and Cityscapes [5] respectively. We present **unconditional generation results** for BAIR in Table 5 and UCF-101 [6] in Table 6, and **interpolation** results for SMMNIST, KTH, and BAIR in Table 7.

**Datasets**: We generate 128x128 images for Cityscapes and 64x64 images for the other datasets. See our Appendix and supplementary material for additional visual results. Our choice of datasets is in order of progressive difficulty: 1) SMMNIST: black-and-white digits; 2) KTH: grayscale single-humans; 3) BAIR: color, multiple objects, simple scene; 4) Cityscapes: color, natural complex natural driving scene; 5) UCF101: color, 101 categories of natural scenes. We process these datasets similarly to prior works. For Cityscapes, each video is center-cropped, then resized to $128 \times 128$. For UCF101, each video clip is center-cropped at 240×240 and resized to 64×64, taking care to maintain the train-test splits.

---

[2] [Denton and Fergus, 2018, Srivastava et al., 2015]   [3] [Schuldt et al., 2004]   [4] [Ebert et al., 2017]   [5] [Cordts et al., 2016]   [6] [Soomro et al., 2012]

Unless otherwise specified, we set the mask probability to 0.5 when masking was used. For sampling, we report results using the sampling methods DDPM [Ho et al., 2020] or DDIM [Song et al., 2020] with only 100 sampling steps, though our models were trained with 1000, to make sampling faster. We observe that the metrics are generally better using DDPM than DDIM (except for UCF-101). Using 1000 sampling steps could yield better results.

Table 1: Video prediction results on SMMNIST ($64 \times 64$) for 10 predicted frames conditioned on 5 past frames. We predicted 10 trajectories per real video, and report the average FVD and maximum SSIM, averaged across 256 test videos.

| SMMNIST [$5 \rightarrow 10$; trained on $k$] | $k$ | FVD↓ | SSIM↑ |
|---|---|---|---|
| SVG [Denton and Fergus, 2018] | 10 | 90.81 | 0.688 |
| vRNN 1L [Castrejón et al., 2019] | 10 | 63.81 | 0.763 |
| Hier-vRNN [Castrejón et al., 2019] | 10 | 57.17 | 0.760 |
| **MCVD** concat (Ours) | **5** | 25.63 | **0.786** |
| **MCVD** spatin (Ours) | **5** | **23.86** | 0.780 |

Note that all our models are trained to predict only 4-5 current frames at a time, unlike other models that predict ≥10. We use these models to then autoregressively predict longer sequences for prediction or generation. This was done in order to fit the models in our GPU memory budget. Despite this disadvantage, we find that our MCVD models perform better than many previous SOTA methods.

**Metrics**: As mentioned earlier, we primarily use the FVD metric for comparison across models as FVD measures both fidelity and diversity of the generated samples. Previous works compare Frechet Inception Distance (FID) [Heusel et al., 2017] and Inception Score (IS) [Salimans et al., 2016], adapted to videos by replacing the Inception network with a 3D-convolutional network that takes video input. FVD is computed similarly to FID, but using an I3D network trained on the huge video dataset Kinetics-400. We also report PSNR and SSIM.

Table 2: Video prediction results on KTH ($64 \times 64$), predicting 30 and 40 frames using models trained to predict $k$ frames at a time. All models condition on 10 past frames, on 256 test videos.

| KTH [$10 \rightarrow pred$; trained on $k$] | $k$ | $pred$ | FVD↓ | PSNR↑ | SSIM↑ |
|---|---|---|---|---|---|
| SAVP [Lee et al., 2018] | 10 | 30 | 374 ± 3 | 26.5 | 0.756 |
| **MCVD** concat (Ours) | **5** | 30 | 323 ± 3 | 27.5 | 0.835 |
| SLAMP [Akan et al., 2021] | 10 | 30 | 228 ± 5 | 29.4 | 0.865 |
| SRVP [Franceschi et al., 2020] | 10 | 30 | 222 ± 3 | **29.7** | **0.870** |
| **MCVD** concat (Ours) | **5** | 40 | 276.7 | 26.40 | 0.812 |
| SAVP-VAE [Lee et al., 2018] | 10 | 40 | 145.7 | 26.00 | 0.806 |
| Grid-keypoints [Gao et al., 2021] | 10 | 40 | **144.2** | **27.11** | **0.837** |

**Ablation studies**: In Table 3 we compare models that use concatenated raw pixels as input to U-Net blocks (concat) to SPATIN variants. We also compare no-masking to past-masking variants, i.e. models which are only trained predict the future vs. models which are regularized by being trained for prediction and unconditional generation. It can be seen that our model works across different choices of past frames and generates better quality for shorter videos. This is expected from models of this kind. Moreover, it can be seen that the model trained on the two tasks of Prediction and Generation (i.e., the models with past-mask) performs better than the model trained only on Prediction!

In addition, the appendix contains an ablation study in Table 9 on the different design choices: concat vs concat past-future-mask vs spatin vs spatin future-mask vs spatin past-future-mask. It can be seen that concat is, in general, better than spatin. It can also be seen that the past-future-mask variant, which is a general model capable of all three tasks, performs better at the individual tasks than the models trained only on the individual task. This was demonstrated in Table 3 as well. This shows that the model gains very helpful insights while generalizing to all three tasks, which it does not while training only on the individual task.

We conducted preliminary experiments with a larger number of frames. Since the models with a larger number of frames were bigger, we could only run them for a shorter time with a smaller batch size than the smaller models. In general, we found that larger models did not substantially improve the results. We attribute this to the fact that using more frames means that the model should be given more capacity, but we could not increase it due to our computational budget constraints. We emphasize that our method works very well with fewer computational resources.

Examining these results we remark that we have SOTA performance for prediction on SMMNIST, BAIR and the challenging Cityscapes evaluation. Our Cityscapes model yields an FVD of 145.5, whereas the best previous result of which we are aware is 418. The quality of our Cityscapes results are illustrated visually in Figure 1 and Figure 2 and in the additional examples provided in our Appendix. While our completely unconditional generation results are strong, we note that when

past masking is used to regularize future predicting models, we see clear performance gains in Table 3. Finally, in Table 7 we see that our interpolation results are SOTA by a wide margin, across experiments on SMMNIST, KTH and BAIR – even compared to architectures much more specialized for interpolation.

It can be seen that our proposed method generates better quality videos, even though it was trained on a shorter number of frames than other methods. It can also be seen that training on multiple tasks using random masking improves the quality of generated frames than training on the individual tasks.

Table 3: Video prediction results on BAIR ($64 \times 64$) conditioning on $p$ past frames and predicting $pred$ frames in the future, using models trained to predict $k$ frames at at time.

| **BAIR** ($64 \times 64$) [past $p \to pred$ ; trained on $k$] | $p$ | $k$ | $pred$ | FVD↓ | PSNR↑ | SSIM↑ |
|---|---|---|---|---|---|---|
| LVT [Rakhimov et al., 2020] | 1 | 15 | 15 | 125.8 | – | – |
| DVD-GAN-FP [Clark et al., 2019] | 1 | 15 | 15 | 109.8 | – | – |
| **MCVD** spatin (Ours) | 1 | **5** | 15 | 103.8 | 18.8 | 0.826 |
| TrIVD-GAN-FP [Luc et al., 2020] | 1 | 15 | 15 | 103.3 | – | – |
| VideoGPT [Yan et al., 2021] | 1 | 15 | 15 | 103.3 | – | – |
| CCVS [Le Moing et al., 2021] | 1 | 15 | 15 | 99.0 | – | – |
| **MCVD** concat (Ours) | 1 | **5** | 15 | 98.8 | 18.8 | 0.829 |
| **MCVD** spatin past-mask (Ours) | 1 | **5** | 15 | 96.5 | 18.8 | 0.828 |
| **MCVD** concat past-mask (Ours) | 1 | **5** | 15 | 95.6 | 18.8 | **0.832** |
| Video Transformer [Weissenborn et al., 2019] | 1 | 15 | 15 | 94-96[a] | – | – |
| FitVid [Babaeizadeh et al., 2021] | 1 | 15 | 15 | 93.6 | – | – |
| **MCVD** concat past-future-mask (Ours) | 1 | **5** | 15 | **89.5** | 16.9 | 0.780 |
| SAVP [Lee et al., 2018] | 2 | 14 | 14 | 116.4 | – | – |
| **MCVD** spatin (Ours) | 2 | **5** | 14 | 94.1 | 19.1 | 0.836 |
| **MCVD** spatin past-mask (Ours) | 2 | **5** | 14 | 90.5 | **19.2** | 0.837 |
| **MCVD** concat (Ours) | 2 | **5** | 14 | 90.5 | 19.1 | 0.834 |
| **MCVD** concat past-future-mask (Ours) | 2 | **5** | 14 | 89.6 | 17.1 | 0.787 |
| **MCVD** concat past-mask (Ours) | 2 | **5** | 14 | **87.9** | 19.1 | **0.838** |
| SAVP [Lee et al., 2018] | 2 | 10 | 28 | 143.4 | – | 0.795 |
| Hier-vRNN [Castrejón et al., 2019] | 2 | 10 | 28 | 143.4 | – | **0.822** |
| **MCVD** spatin (Ours) | 2 | **5** | 28 | 132.1 | 17.5 | 0.779 |
| **MCVD** spatin past-mask (Ours) | 2 | **5** | 28 | 127.9 | 17.7 | 0.789 |
| **MCVD** concat (Ours) | 2 | **5** | 28 | 120.6 | 17.6 | 0.785 |
| **MCVD** concat past-mask (Ours) | 2 | **5** | 28 | 119.0 | **17.7** | 0.797 |
| **MCVD** concat past-future-mask (Ours) | 2 | **5** | 28 | **118.4** | 16.2 | 0.745 |

[a] 94 on only the first frames, 96 on all subsequences of test frames

Table 4: Video prediction on Cityscapes ($128 \times 128$) conditioning on 2 frames and predicting 28. SPATIN seems to produce a drift towards brighter images with a color balance shift in frames further from the start frame on Cityscapes, resulting in increased FVD for SPATIN than the CONCAT variant.

| **Cityscapes** ($128 \times 128$) [$2 \to 28$; trained on $k$] | $k$ | FVD↓ | LPIPS↓ | SSIM↑ |
|---|---|---|---|---|
| SVG-LP Denton and Fergus [2018] | 10 | 1300.26 | $0.549 \pm 0.06$ | $0.574 \pm 0.08$ |
| vRNN 1L Castrejón et al. [2019] | 10 | 682.08 | $0.304 \pm 0.10$ | $0.609 \pm 0.11$ |
| Hier-vRNN Castrejón et al. [2019] | 10 | 567.51 | $0.264 \pm 0.07$ | $0.628 \pm 0.10$ |
| GHVAE Wu et al. [2021] | 10 | 418.00 | $0.193 \pm 0.014$ | **0.740** $\pm 0.04$ |
| **MCVD** spatin past-mask (Ours) | **5** | 184.81 | $0.121 \pm 0.05$ | $0.720 \pm 0.11$ |
| **MCVD** concat past-mask (Ours) | **5** | **141.31** | **0.112** $\pm 0.05$ | $0.690 \pm 0.12$ |

## 5 Conclusion

We have shown how to obtain SOTA video prediction and interpolation results with randomly masked conditional video diffusion models using a relatively simple architecture. We found that past-masking was able to improve performance across all model variants and configurations tested. We believe our approach may pave the way forward toward high quality larger-scale video generation.

**Limitations.** Videos generated by these models are still small compared to real movies, and they can still become blurry or inconsistent when the number of generated frames is very large. Our unconditional generation results on the highly diverse UCF-101 dataset are still far from perfect. More work is clearly needed to scale these

Table 5: Unconditional generation of BAIR video frames.

| **BAIR** ($64 \times 64$) [$0 \to pred$; trained on 5] | $pred$ | FVD↓ |
|---|---|---|
| **MCVD** spatin past-mask (Ours) | 16 | 267.8 |
| **MCVD** concat past-mask (Ours) | 16 | **228.5** |
| **MCVD** spatin past-mask (Ours) | 30 | 399.8 |
| **MCVD** concat past-mask (Ours) | 30 | **348.2** |

models to larger datasets with more diversity and with longer duration video. As has been the case in many other settings, simply using larger models with many more parameters is a strategy that is likely to improve the quality and flexibility of these models – we were limited to 4 GPUs for our work here. There is also a need for faster sampling methods capable of maintaining quality over time.

Given our strong interpolation results, conditional diffusion models which generate skipped frames could make it possible to generate much longer, but consistent video through a strategy of first generating sparse distant frames in a block, followed by an interpolative diffusion step for the missing frames.

Table 6: Unconditional generation of UCF-101 video frames.

| **UCF-101** ($64 \times 64$) [$0 \to 16$; trained on $k$] | $k$ | FVD↓ |
|---|---|---|
| MoCoGAN-MDP [Yushchenko et al., 2019] | 16 | 1277.0 |
| **MCVD** concat past-mask (Ours) | 4 | 1228.3 |
| TGANv2 [Saito et al., 2020] | 16 | 1209.0 |
| **MCVD** spatin past-mask (Ours) | 4 | 1143.0 |
| DIGAN [Yu et al., 2022] | 16 | **655.0** |

Table 7: Video Interpolation results ($64 \times 64$). Given $p$ past + $f$ future frames $\to$ interpolate $k$ frames. Reporting average of the best metrics out of $n$ trajectories per test sample. $\downarrow (p+f)$ and $\uparrow k$ is harder. We used MCVD spatin past-mask for SMMNIST and KTH, and MCVD concat past-future-mask for BAIR. We also include results on SMMNIST for a "pure" model trained without any masking.

| | **SMMNIST** ($64 \times 64$) | | | | | **KTH** ($64 \times 64$) | | | | | **BAIR** ($64 \times 64$) | | | | |
|---|---|---|---|---|---|---|---|---|---|---|---|---|---|---|---|
| | $p+f$ | $k$ | $n$ | PSNR↑ | SSIM↑ | $p+f$ | $k$ | $n$ | PSNR↑ | SSIM↑ | $p+f$ | $k$ | $n$ | PSNR↑ | SSIM↑ |
| SVG-LP Denton and Fergus [2018] | 18 | 7 | 100 | 13.543 | 0.741 | 18 | 7 | 100 | 28.131 | 0.883 | 18 | 7 | 100 | 18.648 | 0.846 |
| FSTN Lu et al. [2017] | 18 | 7 | 100 | 14.730 | 0.765 | 18 | 7 | 100 | 29.431 | 0.899 | 18 | 7 | 100 | 19.908 | 0.850 |
| SepConv Niklaus et al. [2017] | 18 | 7 | 100 | 14.759 | 0.775 | 18 | 7 | 100 | 29.210 | 0.904 | 18 | 7 | 100 | 21.615 | 0.877 |
| SuperSloMo Jiang et al. [2018] | 18 | 7 | 100 | 13.387 | 0.749 | 18 | 7 | 100 | 28.756 | 0.893 | – | – | – | – | – |
| SDVI full Xu et al. [2020] | 18 | 7 | 100 | 16.025 | 0.842 | 18 | 7 | 100 | 29.190 | 0.901 | 18 | 7 | 100 | 21.432 | 0.880 |
| SDVI Xu et al. [2020] | 16 | 7 | 100 | 14.857 | 0.782 | 16 | 7 | 100 | 26.907 | 0.831 | 16 | 7 | 100 | 19.694 | 0.852 |
| **MCVD** (Ours) | **10** | **10** | 100 | 20.944 | 0.854 | **15** | **10** | 100 | 34.669 | 0.943 | **4** | **5** | 100 | 25.162 | 0.932 |
| | **10** | **5** | **10** | 27.693 | 0.941 | **15** | **10** | **10** | 34.068 | 0.942 | **4** | **5** | **10** | 23.408 | 0.914 |
| | | pure | | 18.385 | 0.802 | **10** | **5** | **10** | 35.611 | 0.963 | | | | | |

**Broader Impacts.** High-quality video generation is potentially a powerful technology that could be used by malicious actors for applications such as creating fake video content. Our formulation focuses on capturing the distributions of real video sequences. High-quality video prediction could one day find use in applications such as autonomous vehicles, where the cost of errors could be high. Diffusion methods have shown great promise for covering the modes of real probability distributions. In this context, diffusion-based techniques for generative modelling may be a promising avenue for future research where the ability to capture modes properly is safety critical. Another potential point of impact is the amount of computational resources being spent for these applications involving the high fidelity and voluminous modality of video data. We emphasize the use of limited resources in achieving better or comparable results. Our submission provides evidence for more efficient computation involving fewer GPU hours spent in training time.

## Acknowledgments and Disclosure of Funding

We thank Digital Research Alliance of Canada for the GPUs which were used in this work. Alexia, Vikram thank their wives and cat for their support. We thank CIFAR for support under the AI Chairs program, and NSERC for support under the Discovery grants program, application ID 5018358.

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
