# OpenReview forum: "MCVD - Masked Conditional Video Diffusion for Prediction, Generation, and Interpolation"
_NeurIPS.cc/2022/Conference — NeurIPS 2022 Accept_

### Official Review · Reviewer_h3Ht · 2022-07-12

**Rating:** 6
**Confidence:** 4
**Soundness:** 3 good
**Presentation:** 3 good
**Contribution:** 2 fair

**Summary:**

The authors proposed a DDPM based model for video prediction, called MCVD, based on randomly masking frames for conditioning. The random masking allows the model to be conditioned on past or future frames, enabling the model to perform a variety of tasks such as infilling, prediction and generation. Authors compared the performance of the proposed method in conditional (on BAIR and Cityescapes), inpainting (on SMMNIST, KTH and BAIR) and unconditional setup (on BAIR and UCF-101), demonstrating its capability on generating high quality videos.



**Questions:**

Please check weaknesses.


**Limitations:**

Yes

**Strengths And Weaknesses:**

Strengths And Weaknesses:
===== Strengths
- The paper is well written and is easy to follow.
- The authors compared the proposed method to many other models in conditional and unconditional setups.
- The proposed method is novel (however there are multiple similar parallel works).

===== Weaknesses:
- Limited datasets. Although, authors tested the proposed method on multiple datasets, I find the choice of datasets to be narrow. In both prediction and unconditional generation, the datasets are limited to smaller size dataset (BAIR and UCF) and there is no experiment on larger datasets such as RoboNet or Kinetics. This makes me wonder if the model is heavily overfitting on the training data and there is not much generalization. FitVid paper (Babaeizadeh et al) has an interesting discussion on how overfitted methods can generate realistic high quality videos but are not capable of generating anything new.

- Analysis. There is limited analysis and ablation studies. While the paper shows great and promising results on video tasks, it is not clear how the design choices affect its results. e.g. why the masking should be random for inpainting? In absence of any ablations, I can argue that a fixed set of conditioned frames is a much better defined problem and may result in even higher quality predictions. Table 3 has some for masking strategies but the paper can use more targeted experiments along the same lines. Another angle is computational issues which is mostly ignored in the paper. One major issue in video prediction literature is the huge number of parameters required for getting good results. How is the situation in a score based model? Does a bigger model substantially improves the result or not? Are these models already overfitting? Such analysis can substantially improve the quality of the paper.

- Limited qualitative results. Given the nature of videos, they are hard to compare on printed paper. I highly recommend authors to host predicted videos in an anonymous website and provide comparative videos with other baselines.

---

> ### Author Response · Authors · 2022-08-02
> **Response to h3Ht [1/2]**
>
> We thank the reviewer for taking the time and effort to provide their detailed feedback on our submission! We are happy to note all the positive comments from the reviewers:
>
> > - This paper is well written, clear
> > - Paper is well written and is easy to follow
> > - Proposed method is novel
> > - Combines all three (prediction, generation, interpolation) within one architecture, unlike prior/concurrent works
> > - Validates the versatility of the proposed architecture on several tasks --- unconditional generation, prediction, and interpolation
> > - Shows great and promising results on video tasks
> > - Compared to many other models in conditional and unconditional setups
> > - The proposed approach is more computational efficient than concurrent works in the literature
> > - More efficient blockwise generation mechanism and the ability to interpolate between frames
> > - Shows better performance even though it was trained on fewer frames at a time
> > - Demonstrates its capability on generating high quality videos
>
> ### Response:
>
> We believe the primary criticism can be attributed to our paper not clearly emphasizing a few key contributions. We hope to clarify these contributions and address the reviewers' concerns below.
>
> > Limited datasets. Although, authors tested the proposed method on multiple datasets, I find the choice of datasets to be narrow. In both prediction and unconditional generation, the datasets are limited to smaller size dataset (BAIR and UCF) and there is no experiment on larger datasets such as RoboNet or Kinetics.
>
> That's a great question! We would be more than happy to conduct experiments on much larger scale datasets. However, we choose to start small, investigate the effectiveness of our method, and progressively grow bigger. You can observe this in the order of datasets we choose -
> 1) SMMINST (64x64) : black and white, simple digits moving in constrained spaces
> 2) KTH (64x64) : grayscale, single humans with specific actions
> 3) BAIR (64x64) : color, robots moving in constrained environment, with multiple objects
> 4) CityScapes (128x128) : color, higher resolution, car moving in complex scenes
> 5) UCF101 (64x64) : color, large variety of actions (101!)
>
> As well as the tasks we chose :
> 1. Video Prediction,
> 2. Video Predition + Generation,
> 3. Video Predition + Generation + Interpolation.
>
> Hence, the next logical step is indeed to scale this up! We also want to remind that for all the above experiments, we only used 4 GPUs at a time, and queuing for these GPUs can take many days if not sometimes weeks! We are surely moving towards trying our method on larger datasets. However, in the meantime, we would like to let the research community take advantage of the significant effort we've made so far in showing the merits of our proposed method in all the above datasets and tasks.
>
> > This makes me wonder if the model is heavily overfitting on the training data and there is not much generalization.
>
> Thanks for the great question! We shall make this clearer in the revised manuscript:
>
> All of our metrics are computed on test data that was NEVER seen by the model during training. We do not report any metrics on the training data, we have strong results on never-before-seen data. So our model is definitely not overfitting, and is definitely generalizing quite well!
>
> > FitVid paper (Babaeizadeh et al) has an interesting discussion on how overfitted methods can generate realistic high quality videos but are not capable of generating anything new.
>
> That is an interesting point made in the FitVid paper. However, it is not applicable in our case since our model is not overfitting to the training data in the first place.

---

> > ### Author Response · Authors · 2022-08-02
> > **Response to h3Ht [2/2]**
> >
> > > There is limited analysis and ablation studies.
> > > it is not clear how the design choices affect its results
> >
> > Thanks for pointing this out! We do have ablation studies; we shall make  it much clearer in the revised manuscript, specifically:
> >
> > Table 3 in the paper provides an ablation study varying the number of past frames conditioned on, as well as the length of the final video, and whether we use a Prediction-specific model or a model trained on Prediction+Generation (past-mask). It can be seen that our model works across different choices of past frames and generates better quality for shorter videos. This is expected from models of this kind. Moreover, it can be seen that the model trained on the two tasks of Prediction and Generation (i.e., the models with past-mask) performs better than the model trained only on Prediction!
> >
> > In addition, the appendix contains an ablation study in Table 9 on the different design choices: concat vs concat past-future-mask vs spatin vs spatin future-mask vs spatin past-future-mask. It can be seen that concat is, in general, better than spatin. It can also be seen that the past-future-mask variant, which is a general model capable of all three tasks, performs better at the individual tasks than the models trained only on the individual task. This was demonstrated in Table 3 as well. This shows that the model gains very helpful insights while generalizing to all three tasks, which it does not while training only on the individual task.
> >
> > > why the masking should be random for inpainting?
> >
> > That's a great question! Allow us to clarify:
> >
> > To be clear, the masking is only random during training. The masking is not random at inference-time for the individual task of Interpolation.
> >
> > In general, we found that training using random masking is better than without. See Table 3 for this ablation on Prediction (as mentioned in the above answer). See below for results on pure Interpolation without masking during training, and Table 7 with masking during training.
> >
> > > In absence of any ablations, I can argue that a fixed set of conditioned frames is a much better defined problem and may result in even higher quality predictions.
> >
> > Our experiments showed that models trained only on Interpolation with a fixed set of conditioned frames (SMMNIST psnr: 18.3847, ssim:0.8023) performed worse than models trained with randomly masked conditioned frames (SMMNIST psnr: 20.944, ssim: 0.854). We shall add this to the main paper.
> >
> > As pointed out above, we do have ablation studies in both Table 3 in the main paper as well as Table 9 in the appendix that tackle this very question. They show that a model trained with a fixed set of conditioned frames performs worse than the model trained with randomly masked conditional frames. We shall make this clearer in the revised manuscript.
> >
> > > Another angle is computational issues which is mostly ignored in the paper. One major issue in video prediction literature is the huge number of parameters required for getting good results. How is the situation in a score based model?
> >
> > Thanks for the great question! We humbly submit that we, in fact, emphasize that we were working with limited resources and that all of our great results took much lesser time, memory, GPUs, and GPU hours than many prior/concurrent works. We provide a thorough report of the computational resources used for each of the datasets (number of parameters, CPU memory used, batch size, GPU, GPU memory used, number of training steps, GPU hours) in Table 8 in the appendix.
> >
> > > Does a bigger model substantially improves the result or not?
> >
> > That's a great question! We had conducted preliminary experiments with larger models, but due to our limited computational budget constraints, we could only run them for a shorter time with a smaller batch size than the smaller models. In general, we found that bigger models did not substantially improve the results. We emphasize that our method works very well with fewer computational resources.
> >
> > > Limited qualitative results. Given the nature of videos, they are hard to compare on printed paper. I highly recommend authors to host predicted videos in an anonymous website
> >
> > We hoped you would ask this! We point the reviewer to the HTML page in the supplementary material, which is an anonymized website we created to showcase our great results! Simply download and extract the supplementary material and open the HTML page. It should show you many videos from all our datasets and all the tasks.
> >
> > ---
> >
> > We hope we have sufficiently answered all of the reviewer's comments point by point, and are happy to engage further on more questions! We highly encourage the reviewer to go through the supplementary material, since many of their questions are already answered there. Considering this, we hope the reviewer increases their rating of our paper, and champions it for publication at this conference.

---

### Official Review · Reviewer_8MsY · 2022-07-16

**Rating:** 6
**Confidence:** 5
**Soundness:** 3 good
**Presentation:** 3 good
**Contribution:** 3 good

**Summary:**

This paper introduces the modified U-Net architecture for taking video frame features and the conditional diffusion process for frame-level image generation. Modified U-Net architecture is equipped with conditioning components with non-recurrent architecture fusing frame-level features. The space-time fusing process is conducted in two-step; channel-wise concatenation of frame-level features and space-time adaptive normalization (SPATIN) inspired by SPADE. This non-recurrent and space-time conditioning U-Net architecture is evaluated on the several video generation, prediction, and interpolation datasets. Although the scale of the datasets is restricted, the versatility of the algorithm is validated.

**Questions:**

### The effect of $k$ on the proposed algorithm
The experiments only show the maximum number of frames during training is 4, 5 for video prediction/generation, and 10 for interpolation. Compared to the other model with $k=10, 15$ for training, I guess that the proposed model seems to show better performance on lower $k$ as explained in lines 257~260. How about the smaller or larger number of frames affect the performance of the proposed algorithm?

### Diversity issues in video generation/prediction
What is the diversity of the generated videos? Most video generation works only report fidelity metrics, such as FVD, FID, and IS. However, the fidelity metrics do not show the diversity of the generated videos. The diversity metric introduced in [1] would be good.

[1] Diverse Video Generation using a Gaussian Process Trigger, ICLR 21

### The scale of the dataset
The experiments only contain small-scale datasets. How about applying the proposed algorithm to a larger scale dataset, such as Kinetics, Something-something v2? Is there any bottleneck of the proposed algorithm except the amount of computation? An algorithm with a non-recurrent and sliding window sampling may be weak in this case compared to recurrent counterparts.

### Time coverage of snippets.
Several works typically count the number of frames, not the time span measured in seconds. It would be good to the time span of the video snippets for generation, prediction and interpolation.


**Limitations:**

I’ve mentioned it in the question part.

**Strengths And Weaknesses:**

### Strengths
- The author validates the versatility of the proposed neural diffusion architecture for video generation on several video generation tasks--- unconditional generation, prediction, and interpolation.

### Weaknesses
- The scale of the datasets used in the experiments is small-scale compared to the other generation methods these days.
The disadvantage of non-recurrent and sliding window sampling is not explained. The word ‘non-recurrent’ only appears in the abstract.

---

> ### Author Response · Authors · 2022-08-02
> **Response 1/2**
>
>
> We thank the reviewer for taking the time and effort to provide their detailed feedback on our submission! We are happy to note all the positive comments from the reviewers:
>
> > - This paper is well written, clear
> > - Paper is well written and is easy to follow
> > - Proposed method is novel
> > - Combines all three (prediction, generation, interpolation) within one architecture, unlike prior/concurrent works
> > - Validates the versatility of the proposed architecture on several tasks --- unconditional generation, prediction, and interpolation
> > - Shows great and promising results on video tasks
> > - Compared to many other models in conditional and unconditional setups
> > - The proposed approach is more computational efficient than concurrent works in the literature
> > - More efficient blockwise generation mechanism and the ability to interpolate between frames
> > - Shows better performance even though it was trained on fewer frames at a time
> > - Demonstrates its capability on generating high quality videos
>
> ### Response:
>
> We believe the primary criticism can be attributed to our paper not clearly emphasizing a few key contributions. We hope to clarify these contributions and address the reviewers' concerns below.
>
> > The scale of the datasets used in the experiments is small-scale compared to the other generation methods these days.
> > The experiments only contain small-scale datasets. How about applying the proposed algorithm to a larger scale dataset, such as Kinetics, Something-something v2? Is there any bottleneck of the proposed algorithm except the amount of computation?
>
> That's a great question! We would be more than happy to conduct experiments on much larger scale datasets. However, we choose to start small, investigate the effectiveness of our method, and progressively grow bigger. You can observe this in the order of datasets we choose -
> 1) SMMINST (64x64) : black and white, simple digits moving in constrained spaces
> 2) KTH (64x64) : grayscale, single humans with specific actions
> 3) BAIR (64x64) : color, robots moving in a constrained environment, with multiple objects
> 4) CityScapes (128x128) : color, higher resolution, car moving in complex scenes
> 5) UCF101 (64x64) : color, large variety of actions (101!)
>
> As well as the tasks we chose :
> 1. Video Prediction,
> 2. Video Predition + Generation,
> 3. Video Predition + Generation + Interpolation.
>
> Hence, the next logical step is indeed to scale this up! We also want to remind that for all the above experiments, we only used 4 GPUs at a time, and queuing for these GPUs can take many days, if not sometimes weeks! We are surely moving towards trying our method on larger datasets. However, in the meantime, we would like to let the research community take advantage of the significant effort we've made so far in showing the merits of our proposed method in all the above datasets and tasks.
>
> > The disadvantage of non-recurrent and sliding window sampling is not explained. The word ‘non-recurrent’ only appears in the abstract.
> > An algorithm with a non-recurrent and sliding window sampling may be weak in this case (scaling to larger datasets) compared to recurrent counterparts.
>
> That's a great question! We shall include the following explanation in the revised manuscript:
>
> Previous video prediction models were typically either 1) non-recurrent: predicting all $n$ frames simultaneously with no way of adding more frames (most GAN-based methods), or 2) recurrent in nature, predicting 1 frame at a time in an autoregressive fashion. The benefit of the non-recurrent type is that you can generate videos faster than 1 frame at a time while still allowing for generating as many frames as needed. The disadvantage is that it is slower than generating all frames at once and takes up more memory and compute at each iteration. Our model finds a sweet spot in between in that it is block-autoregressive: generating $k<n$ frames at a time recurrently to finally obtain $n$ frames.
>
> > the proposed model seems to show better performance on lower $k$ as explained in lines 257~260.
>
> That's exactly right! The proposed method generates better quality videos, even though it was trained on a shorter number of frames than other methods.

---

> > ### Author Response · Authors · 2022-08-02
> > **Response 2/2**
> >
> > > How about the smaller or larger number of frames affect the performance of the proposed algorithm?
> >
> > That's a great question! We shall mention the following in the revised manuscript:
> >
> > We had conducted preliminary experiments with a larger number of frames. Since the models with a larger number of frames were bigger, we could only run them for a shorter time with a smaller batch size than the smaller models. In general, we found that larger models did not substantially improve the results. We attribute this to the fact that using more frames means that the model should be given more capacity, but we could not increase it due to our computational budget constraints. We emphasize that our method works very well with fewer computational resources.
> >
> > > What is the diversity of the generated videos? Most works report fidelity metrics, such as FVD, FID, and IS.
> >
> > That's a great question! It has been shown in [1, 2] that FID is a good measure of BOTH fidelity and diversity, while IS is a good measure of fidelity but not diversity. It has been shown that the properties of FID transfer to FVD [3], since they are both computed using the same principles. Hence, we considered FVD as an effective indicator of diversity.
> >
> > [1] Large Scale GAN Training for High Fidelity Natual Image Synthesis
> > [2] An empirical study on evaluation metrics of generative adversarial networks
> > [3] Towards Accurate Generative Models of Video: A New Metric & Challenges
> >
> > > The diversity metric introduced in [1] would be good. [1] Diverse Video Generation using a Gaussian Process Trigger, ICLR 21
> >
> > Thank you for the suggestion! As mentioned above, FVD itself is a good measure of diversity and is a standard metric reported by several prior works. Hence, we use FVD itself as the diversity metric, and show that our videos are indeed quite diverse. This can be qualitatively checked in the video samples in the appendix.
> >
> > > Time coverage of snippets.
> > > Several works typically count the number of frames, not the time span measured in seconds. It would be good to the time span of the video snippets for generation, prediction and interpolation.
> >
> > That's an interesting point. The time span of each video depends on the dataset; different datasets have been recorded at different frame rates. However, an advantage of our proposed method is that its effectiveness does not rely on frame rate; it can interpolate frames to even increase the frame rate. So the number we chose to report is the number of frames and not the time span.
> >
> > ---
> >
> > We hope we have sufficiently answered all of the reviewer's comments point by point, and are happy to engage further on more questions! Considering this, we hope the reviewer increases their rating of our paper, and champions it for publication at this conference.

---

> > > ### Comment · Reviewer_8MsY · 2022-08-09
> > > **Thank you for the response**
> > >
> > > I've read the response from the author and most of my questions are solved.
> > >
> > > The proposed method seems to have marginal contributions over the previous works. However, the presented materials are straightforward to understand and the proposed framework for integrated video generation-related tasks is novel and promising for future research direction.
> > > I hope that the final version of the paper includes more analysis and discussion, such as more evaluation metrics, characteristics of used datasets, and different conditional frames for training, than the current version because the experiments are conducted on a relatively small scale.
> > >
> > > Thank you.

---

### Official Review · Reviewer_LFuX · 2022-07-17

**Rating:** 5
**Confidence:** 5
**Soundness:** 3 good
**Presentation:** 3 good
**Contribution:** 3 good

**Summary:**

This work tackles the tasks of video generation, prediction and interpolation using a score-based diffusion modeling approach. The main innovation is in using diffusion models to block-wise autoregressively generate/predict/interpolate video both forward and backward in time efficiently (i.e. using <4GPUs in <12days).

**Questions:**

BAIR, KTH, UCF-101 are some of the standard video generation benchmarks. Although the results tables appear to be quite exhaustive for BAIR/KTH, UCF101 is lacking in comparison... which matters since it is the standard benchmark for unconditional video generation. Could you improve the UCF-101 benchmark by making the experimental procedure clearer (see Appendix A.2 in the [DIGAN](https://openreview.net/pdf?id=Czsdv-S4-w9) paper), also do provide FID and IS results on this dataset, appropriately presented to allow for easy comparison to prior work such as progressiveVGAN,LDVD-GAN/TGAN-F,TGAN-ODE, etc.


The introductory paragraph touching on vehicle safety is an interesting way to introduce the problem, but it isn't touched upon beyond the first paragraph. In the end, how does your work impact this?


**Limitations:**

More effort and thought has to be put into the broader impact statement. Two generic sentences on broader impact is unacceptable for this kind of work.

**Strengths And Weaknesses:**

Strengths:
1. This paper is well written, clear and provides a straight-forward extension of the diffusion modelling approach to video.
2. The proposed approach is more computational efficient than concurrent works in the literature (i.e. using <4GPUs in <12days).
3. Technical contributions include a more efficient blockwise generation mechanism and the ability to interpolate between frames. Concurrent works use similar architectures and are capable of video generation and prediction. This work combines all of these within one architecture.

Weaknesess:
1. This work misses comparisons to prior video GAN based work. Taking the UCF-101 experiments as an example, although this work motivates the case for focusing on FVD as a metric, prior work utilises FID and IS as metrics. Not evaluating on these metrics makes it difficult to compare this work against prior art. For historical comparison, I would strongly encourage the authors to extend or append their results to a table similar to the one found in other concurrent/prior works (e.g. Table 1 from the [Video Diffusion Models](https://arxiv.org/abs/2204.03458) or [DIGAN](https://openreview.net/forum?id=Czsdv-S4-w9) papers)
2. This work is limited to 64x64 generation. Many GAN based approaches are more performant and efficient at this resolution. The 64x64 generation should be made explicitely clear in any results/discussions, especially when comparing results to other works. [TGAN-F/LDVD-GAN](https://www.sciencedirect.com/science/article/abs/pii/S0893608020303397) discussed how results from these metrics are dependant on resolution and the pre-processing pipeline.
3. Although an interesting direction for diffusion models, the novelty of this work in terms of new capabilities is limited when compared to prior art or GAN-based approaches (e.g.  [Deep Video Generation, Prediction and Completion of Human Action Sequences - CVPR 2018](https://arxiv.org/abs/1711.08682)).

---

> ### Author Response · Authors · 2022-08-02
> **Response 1/2**
>
> We thank the reviewer for taking the time and effort to provide their detailed feedback on our submission! We are happy to note all the positive comments from the reviewers:
>
> > - This paper is well written, clear
> > - Paper is well written and is easy to follow
> > - Proposed method is novel
> > - Combines all three (prediction, generation, interpolation) within one architecture, unlike prior/concurrent works
> > - Validates the versatility of the proposed architecture on several tasks --- unconditional generation, prediction, and interpolation
> > - Shows great and promising results on video tasks
> > - Compared to many other models in conditional and unconditional setups
> > - The proposed approach is more computational efficient than concurrent works in the literature
> > - More efficient blockwise generation mechanism and the ability to interpolate between frames
> > - Shows better performance even though it was trained on fewer frames at a time
> > - Demonstrates its capability on generating high quality videos
>
> ### Response:
>
> We believe the primary criticism can be attributed to our paper not clearly emphasizing a few key contributions. We hope to clarify these contributions and address the reviewers' concerns below.
>
> > misses comparisons to prior video GAN based work
>
> We address specific points below.
>
>
> > prior work utilises FID and IS as metrics
>
> We initially did consider FID and IS for comparison, however, since FID and IS are image-based metrics while FVD is video-based, we opted to measure FVD. We believe that even if FID is computed using a feature extractor that takes video input, FVD is also computed precisely in that way, using an I3D network trained on the huge video dataset Kinetics-400. We refer to the FVD paper [1] for more details on FVD.
>
> Nevertheless, we are working on computing FID and IS. We aim to have these results by the end of the discussion period. We will include them in a response and in the revised manuscript (along with further comparisons to other work which only computed FID/IS).
>
> [1] FVD: A new metric for video generation; workshop paper at ICLR 2019 https://openreview.net/pdf?id=rylgEULtdN
>
>
>
> > limited to 64x64 generation, many GAN based approaches are more performant and efficient at this resolution
>
> We actually use 128x128 for Cityscapes and 64x64 for all other datasets.
>
> We ask the reviewer not to ignore that score-based denoising diffusion models are very different from GANs. GANs have their own advantages and disadvantages; for example, GANs are very hard to train and have trouble with generating diverse data compared to denoising diffusion models [1, 2, 3, 4]. Even if GAN-based approaches are more efficient at this resolution, we don't see that as a weakness of our submission.
>
> [1] Tackling the generative learning trilemma with denoising diffusion gans, https://arxiv.org/abs/2112.07804
> [2] Pacgan: The power of two samples in generative adversarial networks
> [3] Towards principled methods for training generative adversarial networks
> [4] A closer look at the optimization landscapes of generative adversarial networks
>
>
> > The 64x64 generation should be made explicitly clear in any results/discussions
>
> Thank you for the great suggestion! We currently mention the resolution in the appendix and the processing pipeline in the code, both of which are provided in the supplementary. However, we agree that it should be stated in the main paper and we will add this information into the manuscript. We actually use 128x128 for Cityscapes and 64x64 for all other datasets.

---

> > ### Author Response · Authors · 2022-08-02
> > **Response 2/2**
> >
> > > novelty is limited when compared to prior art or GAN-based approaches (e.g. Deep Video Generation, Prediction and Completion of Human Action Sequences - CVPR 2018)
> >
> > We respectfully emphasize that our work is not similar to the mentioned paper (see below for the differences). Our approach is more general compared to the cited work (it does not rely on specialized skeleton generation as an intermediate representation) and provides new capabilities in that sense (it works on arbitrary video, not just humans). Moreover, the mentioned paper requires additional optimization while ours does not.
> >
> > Our novelty is in using score-based diffusion models to solve all three video tasks (prediction, generation, interpolation) simultaneously. We ask the reviewer not to ignore the fact that diffusion models are very different from GANs in many respects.
> >
> > | Deep Video Generation, Prediction and Completion of Human Action Sequences - CVPR 2018 | Ours |
> > | -------- | -------- |
> > | Uses a GAN. | Uses conditional score-based denoising diffusion models. |
> > | Shows results only on 1 dataset: Human3.6M. | Shows results on 5 datasets: SMMINST, KTH, BAIR, CityScapes, UCF101. |
> > | Subtracts all the backgrounds, and focuses on the foreground human figure only. | Generates full image of complex scenes. |
> > | First generates human pose sequences, then transfers them onto human bodies in pixel space. | Does not have an intermediate stage of human pose sequenes, thus is not constrained to human-only videos. |
> > | Performs prediction and interpolation using randomized optimization on the latent space. | Does not need any further optimization, the same single model is capable of all three tasks simultaneously (as mentioned by the reviewer). |
> >
> >
> >
> > > Could you improve the UCF-101 benchmark by making the experimental procedure clearer
> >
> > Thank you for the great suggestion! We followed the same procedure as mentioned in Appendix A2 in the DIGAN paper. We shall mention this explicitly in the revised manuscript.
> >
> >
> >
> > > provide FID and IS results on this dataset, appropriately presented to allow for easy comparison to prior work such as progressiveVGAN,LDVD-GAN/TGAN-F,TGAN-ODE, etc.
> >
> > As mentioned earlier, we are working on computing FID and IS. We aim to have these results by the end of the discussion period. We will include them in a response and in the revised manuscript (along with further comparisons).
> >
> > > Two generic sentences on broader impact is unacceptable
> >
> > Thank you for bringing this to our notice. We plan to mention the following as part of the broader impact:
> >
> > *High-quality video generation is potentially a powerful technology that could be used by malicious actors for applications such as creating fake video content. Our formulation here, however, focuses on capturing the distributions of real video sequences and not on modifying or altering video. High-quality video prediction could one day find use in applications such as autonomous vehicles, where the cost of errors could be high. Interestingly, diffusion methods have shown great promise for covering the modes of real probability distributions compared to other generative modelling techniques. In this context, diffusion-based techniques for generative modelling may be a promising avenue for future research where the ability to capture modes properly is safety critical.*
> >
> > *Another potential point of impact is the amount of computational resources being spent for these applications involving the high fidelity and voluminous modality of video data. We emphasize the use of limited resources in achieving better or comparable results. Our submission provides evidence for more efficient computation involving fewer GPU hours spent in training time.*
> >
> > If the reviewer feels that other key issues should be discussed, we are happy to include additional statements and discuss any other issues as appropriate.
> >
> > ---
> >
> > We hope we have sufficiently answered all of the reviewer's comments point by point and are happy to engage further on more questions! Considering this, we hope the reviewer increases their rating of our paper.

---

> > > ### Comment · Reviewer_LFuX · 2022-08-08
> > > **Good Response**
> > >
> > > I look forward to reading the revised manuscript.
> > >
> > > In particular, a comprehensive table comparing FID&IS results to all prior art on the associated benchmark datasets.
> > >
> > > I also expect it to be made clear, when working on 64x64 vs 128x128 video generation.
> > >
> > > The comparison to "Deep Video Generation, Prediction and Completion of Human Action Sequences - CVPR 2018" was very insightful.
> > >
> > > Regarding the impact statement.. it is inaccurate to say that you do not alter video when interpolation does just that.
> > > please consider rephrasing the second sentence to something like:
> > >
> > > __"Our formulation focuses on capturing the distributions of real video sequences."__
> > >
> > >
> > > If the authors address all my other concerns (and those raised by other reviewers) in their updated manuscript, i am willing to further reflect that in my review score.

---

### Author Response · Authors · 2022-08-02
**Thank you!**

We thank all the reviewers for taking the time and effort to provide their detailed feedback on our submission! We have taken note of every comment, and all of them have contributed to making our submission even better. We are happy to note all the positive comments from the reviewers:

> - This paper is well written, clear
> - Paper is well written and is easy to follow
> - Proposed method is novel
> - Combines all three (prediction, generation, interpolation) within one architecture, unlike prior/concurrent works
> - Validates the versatility of the proposed architecture on several tasks --- unconditional generation, prediction, and interpolation
> - Shows great and promising results on video tasks
> - Compared to many other models in conditional and unconditional setups
> - The proposed approach is more computational efficient than concurrent works in the literature
> - More efficient blockwise generation mechanism and the ability to interpolate between frames
> - Shows better performance even though it was trained on fewer frames at a time
> - Demonstrates its capability on generating high quality videos

We believe the primary criticism can be attributed to our paper not clearly emphasizing a few key contributions. We hope to clarify these contributions and address the reviewers' concerns below. We had already provided many comparisons to many other papers for multiple datasets. Nevertheless, we will now also provide the FID and IS metrics (since some papers only report on the FID and IS, and not FVD) as requested by reviewer LFuX.

We recommend the reviewers also check our appendix in the supplementary material. We provided a detailed report in Table 8 on the computational resources used for each of the datasets: number of parameters, CPU memory used, batch size, GPU, GPU memory used, number of training steps, and GPU hours. We also provided an additional ablation study in Table 9, as well as some qualitative examples.

We also refer the reviewers to our qualitative results presented in an HTML file in the supplementary material. It contains several videos generated using our model, showcasing its effectiveness in all three tasks of Video Prediction, Generation, and Interpolation.

---

> ### Comment · Reviewer_LFuX · 2022-08-09
> **revised manuscript with changes highlighted**
>
> Could you provide a revised manuscript with all the changes made highlighted in a different color?

---

### Author Response · Authors · 2022-08-09
**IS and inception results**

As asked by the reviewers, we updated the paper with the revisions and highlighted the changes.

Note that we do not yet have the IS and FID results because we were delayed due to problems with the chainer and cupy packages dependencies and we had to contact the authors for further help. We will post the results once we obtain them if this is allowed by the openreview system, otherwise they will be included in the camera-ready version.

---

### Meta-Review · Area_Chair_Rnnw · 2022-08-29

**Recommendation:** Accept
**Confidence:** Certain

**Metareview:**

The paper proposes the use of diffusion model for masked video modeling and shows promising results in video generation and completion. All of the reviewers agree that the paper is a good fit for publication at NeurIPS. I appreciate that the authors engaged with the reviewers and improved the paper!

**Award:**

No

---

### Decision · Program_Chairs · 2022-09-14

Accept